# Hector V3.2.0: functionality and performance of a reduced-complexity climate model

**Kalyn Dorheim**[1], **Skylar Gering**[2], **Robert Gieseke**[3], **Corinne Hartin**[4], **Leeya Pressburger**[1], **Alexey N. Shiklomanov**[5], **Steven J. Smith**[1], **Claudia Tebaldi**[1], **Dawn L. Woodard**[1], and **Ben Bond-Lamberty**[1]

[1]Joint Global Change Research Institute, Pacific Northwest National Laboratory,
5825 University Research Ct, Suite 3500, College Park, Maryland 20740, USA
[2]California Institute of Technology, 1200 East California Boulevard, Pasadena, California 91125, USA
[3]independent researcher: Potsdam, Germany
[4]Climate Change Division, Office of Atmospheric Protection, U.S. Environmental Protection
Agency, Washington, DC 20004, USA
[5]NASA Goddard Space Flight Center, 8800 Greenbelt Rd, Greenbelt, Maryland 20771, USA

**Correspondence:** Kalyn Dorheim (kalyn.dorheim@pnnl.gov)

**Abstract.** Hector is an open-source reduced-complexity climate–carbon cycle model that models critical Earth system processes on a global and annual basis. Here, we present an updated version of the model, Hector V3.2.0 (hereafter Hector V3), and document its new features, implementation of new science, and performance. Significant new features include permafrost thaw, a reworked energy balance submodel, and updated parameterizations throughout. Hector V3 results are in good general agreement with historical observations of atmospheric $CO_2$ concentrations and global mean surface temperature, and the future temperature projections from Hector V3 are consistent with more complex Earth system model output data from the sixth phase of the Coupled Model Intercomparison Project. We show that Hector V3 is a flexible, performant, robust, and fully open-source simulator of global climate changes. We also note its limitations and discuss future areas for improvement and research with respect to the model's scientific, stakeholder, and educational priorities.

## 1 Introduction

Reduced-complexity climate models (RCMs) play a critical role within the diverse climate modeling landscape (Sarofim et al., 2021). Using strategically simpler representations of large-scale climate processes and dynamics compared to coupled Earth system models (ESMs), RCMs are computationally efficient sources of future climate projections; they are able to produce large ensembles of results and explore key uncertainties at a fraction of the computational cost of a single ESM run (Kawamiya et al., 2020). For this reason, RCMs such as Hector, the Model for the Assessment of Greenhouse Gas Induced Climate Change (MAGICC), the Finite-amplitude Impulse Response (FaIR) model, and the other models participating in the Reduced Complexity Model Intercomparison Project (RCMIP) (Nicholls et al., 2021; Meinshausen et al., 2011; Smith et al., 2018; Nicholls et al., 2020) have been coupled with socioeconomic models (Calvin et al., 2019); been used to study climate–carbon interactions and feedbacks (Woodard et al., 2021); supported the assessment of key quantities, such as global temperature and the carbon budget, in various Intergovernmental Panel on Climate Change (IPCC) reports (Smith et al., 2021; Forster et al., 2021); and served other applications.

Hector is a globally resolved carbon–climate RCM with explicit terrestrial and ocean carbon cycles as well as active surface ocean chemistry. As a stand-alone climate model, Hector has been used in a variety of other research projects (Woodard et al., 2021; Dorheim et al., 2020; Schwarber et al., 2019; Vega-Westhoff et al., 2019; Pressburger et al., 2023) and participated in the first two phases of the RCMIP (Nicholls et al., 2021, 2020). Additionally, since 2015, Hector has been the climate component of the Global Change Analysis Model (GCAM) (Calvin et al., 2019) and has been used to explore the feedback from hydrofluorocarbon emissions from future changes in terms of heating and cooling degree days (Hartin et al., 2021) as well as how carbon dioxide ($CO_2$) removal technologies may impact the energy–water–land system (Fuhrman et al., 2023).

Since the initial release of Hector, its model development has continued in order to reflect the advances made within the communities of climate science and open-source software research, and the objective of this paper is to document the latest version of the model. We provide an overview of the model before describing the major changes and upgrades that have been made since Hector V1, focusing on the default model configuration as well as describing optional settings. We then compare Hector V3 results with observations and ESM output to examine model performance. Finally, we discuss future areas for improvement for the model in the context of its goals of accuracy, performance, and broad accessibility.

## 2 Methods

### 2.1 General description of the model

The first version of Hector (V1) was described in detail by Hartin et al. (2015). It is a self-contained object-oriented model implemented in C++ with a flexible, modular design. While Hector produces annual output, its adaptive time solver is capable of operating at a higher frequency to help address issues with numerical instability.

In Hector's default configuration, all model runs begin after "spin-up" (Thornton and Rosenbloom, 2005), during which the model runs until all carbon pools are in equilibrium; this typically requires $\sim 300$ years using the default model parametrization and typically results in changes of a few percent in the model's major carbon pools. After the spinup phase is complete, the main Hector run begins. A Hector run can either be "free running" or "constrained". By default, the model is free running, meaning that its behavior is determined by the time series of emissions and other inputs. During a constrained run, the model is forced to match one or more user-prescribed time series. The default free-running model uses time series from 37 different emission species and three exogenous radiative forcers (see Table S1). These emission inputs fall into two categories. The first category consists of emissions that accumulate as greenhouse gas (GHG) concentrations. The GHG concentrations for nitrous oxide ($N_2O$), methane ($CH_4$), and 26 halocarbons are calculated using equations that encode a simplified relationship between emissions and concentrations (Tables S3–S5). The GHG concentrations for ozone ($O_3$) are calculated from interactions between nitrogen oxides ($NO_x$), carbon monoxide (CO), and non-methane volatile organic compound (NMVOC) emissions (Eqs. S42–S43 in Table S10). The atmospheric $CO_2$ concentrations are determined in part by anthropogenic $CO_2$ emissions (read in as an input) and by the behavior of Hector's terrestrial and ocean carbon cycle components (Fig. 1). The second category consists of the emissions that impact Hector's radiative-forcing budget: carbon monoxide (CO), black carbon (BC), organic carbon (OC), sulfur dioxide ($SO_2$), and ammonia ($NH_3$) emissions. These emissions are used in Eqs. S12–S16, which determine aerosol concentrations and thus radiative forcings. Total radiative forcing is the sum of the forcing effects of all of Hector's atmospheric greenhouse gases, all aerosols, and several additional forcing inputs (volcanic forcing, albedo).

Total radiative forcing is then used to simulate temperature change. Hector's temperature component (Vega-Westhoff et al., 2019) is an implementation of the Diffusion Ocean Energy balance CLIMate (DOECLIM) model (Kriegler, 2005; Tanaka et al., 2007). The DOECLIM model is a 1-D purediffusion ocean model that calculates changes in air temperature 2 m over ocean/land, changes in sea surface temperature, and changes within the ocean mixed layer. The sea surface and land surface temperatures from the DOECLIM model are used by Hector's ocean and land carbon cycles to calculate the carbon fluxes at the next time step. Hector's global mean surface temperature (GMST) is the area-weighted average of these land surface and ocean surface temperatures.

### 2.2 Changes since V1

A number of significant architectural, software, and scientific developments have been implemented since the release of V1 and its documentation (Hartin et al., 2015). We start by documenting these software changes before discussing other changes and new features affecting Hector's carbon cycle, radiative forcing, temperature calculations, and constrained-mode capabilities.

#### 2.2.1 Software

Hector is an open-source community model available on GitHub (https://github.com/jgcri/hector, last access: 29 May 2024). The repository includes updated project solutions and makefiles to support the building and running of Hector from the command line or within development environments like Visual Studio (https://visualstudio.microsoft.com/, last access: 29 May 2024) or Xcode (https://developer.apple.com/xcode/, last access: 29 May 2024). Alternatively, users can

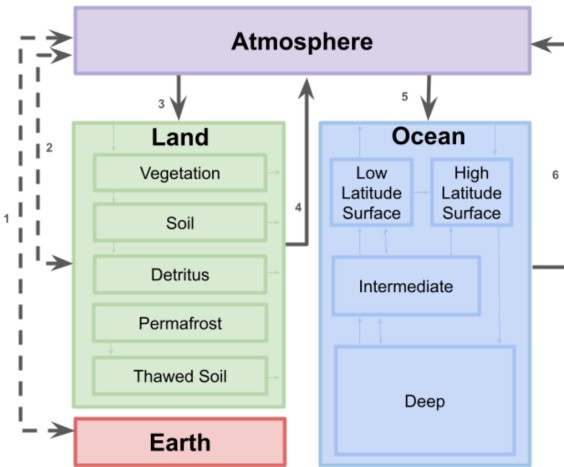

**Figure 1.** Conceptual diagram of the $CO_2$ fluxes (thick gray arrows labeled with numbers) between Hector's four major carbon cycle boxes: a well-mixed atmosphere ("Atmosphere"), a terrestrial carbon cycle ("Land"), an ocean carbon cycle ("Ocean"), and a geological fossil fuel reservoir ("Earth"). The thinner arrows within the "Land" and "Ocean" boxes allude to Hector's more complex submodule carbon cycle dynamics, which are not discussed in detail here. The solid lines indicate that $CO_2$ fluxes are calculated within Hector, whereas the dashed lines indicate that the fluxes are externally defined inputs that are read into the model. The two-headed arrows indicate a potential two-way exchange of carbon. The fluxes are as follows: (1) $CO_2$ emissions from fossil fuels and industry and uptake of carbon capture technologies, (2) $CO_2$ emissions and uptake from land use change (afforestation, deforestation, etc.), (3) vegetation uptake from the atmosphere, (4) aggregate $CO_2$ from respiration from the terrestrial biosphere, (5) ocean carbon uptake, and (6) outgassing. The model's permafrost implementation (Woodard et al., 2021) emits both $CO_2$ and $CH_4$ into the atmosphere from its "Thawed Soil" pool, whereas the "Soil" pool emits only heterotrophic $CO_2$ respiration.

run Hector as an R package (R Core Team, 2021), allowing for a broader range of users given R's popularity as a data analysis and simulation tool across many scientific disciplines. The R package wrapper enabled the development of the Hector user interface (UI) (Pennington and Vernon, 2021), which allows users to run and interact with Hector results in a web browser. Other changes include updated and reduced software dependencies, automated software testing, and auto-generated online documentation. Finally, a Python wrapper, pyhector (Willner et al., 2017), is maintained by community collaborators, broadening the range of potential users and use cases of the model. The default model remains highly performant: even without any speed optimizations at compile time, running the 550 years (1750–2300) of a standard run takes $\sim 0.5\,\mathrm{s}$ on a modern laptop. The model is also straightforward to parallelize for large-ensemble analyses (Pressburger et al., 2023). Ultimately, these Hector V3 software changes have led to a more robust, transparent, and accessible community model.

### 2.2.2 Carbon cycle

Anthropogenic $CO_2$ emissions are debited from a geological pool (named "earth" in Hector; see Fig. 1) and added to the one-pool, global atmosphere at each time step. Hector's active carbon cycle is split into terrestrial land and ocean submodels.

As described in detail by Hartin et al. (2015, 2016), Hector's ocean carbon cycle is a four-box module consisting of two surface-level ocean boxes, an intermediate ocean box, and a deep ocean box (Fig. 1). Carbon and water mass exchanges occur between the four boxes, respecting simplified representations of advection and thermohaline circulation, with volume transports tuned to approximate a flow of $100\,\mathrm{Pg}\,\mathrm{C}$ from the surface-level high-latitude box to the deep ocean box at a steady state, simulating deep-water formation. Hector solves for the marine carbonate variables – dissolved inorganic carbon (DIC), pH, and alkalinity – with respect to solubility in the two surface-level boxes (Zeebe and Wolf-Gladrow, 2001). The calculation of $pCO_2$ in each surface box is based on the concentration of $CO_2$ in the ocean and its solubility, which is in turn a function of temperature, salinity, and pressure. At a steady state, the cold high-latitude surface box ($> 55°\,\mathrm{N}$ or S) acts as a sink of carbon from the atmosphere, while the warm low-latitude surface box ($\leq 55°\,\mathrm{N}$ or S) off-gases carbon back to the atmosphere. The ocean–atmosphere flux calculation follows Takahashi et al. (2009). In Hector V3, ocean carbon cycle calculations use sea surface temperature (SST) calculated by the DOECLIM model (see above), and the preindustrial surface-level, intermediate, and deep ocean carbon cycle pools are initialized from the IPCC Sixth Assessment Report (AR6) – Fig. 5.12 of Canadell et al., 2021 (see Table 1).

Much of the basic functionality of the model's terrestrial carbon cycle remains unchanged from the original V1 release (Hartin et al., 2015). Net primary production (NPP) is partitioned into vegetation, detritus, and soil (Fig. 1); litterfall moves carbon from vegetation to the soil, and temperature-dependent, first-order decay equations control the heterotrophic release of $CO_2$ back to the atmosphere from the latter two pools (Hartin et al., 2015). By default, the terrestrial carbon cycle operates as a single global biome, but Hector can run with an arbitrary number of independent biomes, each with its own set of carbon pools and parameters; a sample multibiome parameterization is included with the model's input files, and an example of this was documented in detail by Woodard et al. (2021).

There are also new or changed behaviors in the Hector V3 terrestrial carbon submodel. Initially, land use change (LUC) emissions were specified as a single time series that could be positive or negative, reflecting net emissions or uptake, and this value was added to (subtracted from) the atmosphere and subtracted from (added to) the vegetation, detritus, and soil pools (Hartin et al., 2015). In V3, these are now provided in separate input time series that must be strictly positive and

**Table 1.** Default Hector parameter values and their sources. The parameter name column shows the names as they appear in the model's INI (initialization) files. This table does not list all Hector parameters but rather contains the parameters that have been updated since Hartin et al. (2015). For a complete collection of parameter values and their sources, refer to the default initialization files available at https://github.com/JGCRI/hector/tree/main/inst/input (last access: 29 May 2024). Preindustrial values here are assumed to be circa 1745, the start of a Hector run.

| Parameter | Description | Value | Units | Source |
| --- | --- | --- | --- | --- |
| CH4N | Natural $CH_4$ emissions are assumed to be constant over the historical and future periods | 338 | $Tg\,CH_4\,yr^{-1}$ | See Sect. 2.2.6 for details |
| N2ON | Natural $N_2O$ emissions are assumed to be constant over the historical and future periods | 9.7 | $Tg\,N\,yr^{-1}$ | |
| beta | $CO_2$ fertilization factor ($\beta$) – increase in NPP productivity with increasing $CO_2$ concentrations | 0.55 | unitless | |
| q10_rh | Heterotrophic respiration temperature sensitivity factor ($Q_{10}$) | 2.2 | unitless | |
| diff | Vertical ocean heat diffusivity ($\kappa$) – the rate at which heat diffuses into the ocean | 1.16 | $cm^2\,s^{-1}$ | |
| preind_surface_c | Initial size of the preindustrial surface-level ocean carbon pool | 900 | Pg C | Fig. 5.12 (Canadell et al., 2021) |
| preind_interdeep_c | Initial size of the preindustrial intermediate and deep ocean carbon pools | 37 100 | Pg C | |
| C0 | Preindustrial $CO_2$ concentration | 277.15 | $ppmv\,CO_2$ | Table 7.SM.1 (Smith et al., 2021) |
| N0 | Preindustrial $N_2O$ concentration | 273.87 | $ppbv\,N_2O$ | |
| M0 | Preindustrial $CH_4$ concentration | 731.41 | $ppbv\,CH_4$ | |
| npp_flux0 | Preindustrial net primary production | 56.2 | $Pg\,C\,yr^{-1}$ | Ito (2011) |
| TOS0 | Mean preindustrial absolute ocean air temperature | 18 | °C | Processed data from the sixth phase of the Coupled Model Intercomparison Project (CMIP6) (Pressburger and Dorheim, 2022) |
| deltaHL0 | Difference between high-latitude preindustrial ocean temperature and TOS0 | −16.4 | °C | |
| deltaLL0 | Difference between low-latitude preindustrial ocean temperature and TOS0 | 2.9 | °C | |

correspond to the gross emissions and uptake fluxes; because of how LUC now affects NPP (see below), they are assumed to include any regrowth fluxes from previous LUC. A similar change has been made to the fossil fuel emissions and industrial emissions, which are now specified by two gross fluxes of emissions and uptake. This provides users with more flexibility to specify how the gross fluxes result in the net flux,

with no behavior change otherwise. Note that the model still accepts net fluxes if that is all that is available, as is the case for the RCMIP Shared Socioeconomic Pathway (SSP) scenarios (Nicholls et al., 2020).

Second, LUC fluxes now affect land carbon pools in proportion to the sizes of these pools, rather than via fixed allocation fractions as previously. This is a more conservative as-

sumption than the previous user-defined allocation approach, given the large uncertainty about LUC flux magnitudes and interactive carbon cycle effects (Yue et al., 2020; Friedlingstein et al., 2023). In addition, in a non-spatial model such as Hector, the carbon pool sizes are governed by the total amount of carbon in the system and the first-order equations linking the pools; LUC loss is only temporary until the pools re-equilibrate. The new approach is thus simpler and, in most cases, will have only minor effects on model results.

Third, terrestrial NPP is now affected by LUC. The model tracks how much cumulative carbon has been lost (or gained) due to LUC, relative to preindustrial conditions, and then adjusts the NPP by this fraction in addition to the pre-existing temperature and $CO_2$ adjustments to NPP described by Hartin et al. (2015). The logic behind this change is that extensive historical deforestation is known to affect photosynthesis and NPP (Ito, 2011; Malhi et al., 2004; Kaplan et al., 2012), and in previous versions of Hector, deforestation did not affect the model's NPP at all. The new behavior is given as

$$\text{NPP}(t) = \text{NPP}_0 \times f(C_{\text{atm}}\beta) \times f(\text{LUC}_v), \tag{1}$$

where $t$ is the current time step, $\text{NPP}_0$ is preindustrial NPP, and the two $f$ terms represent $CO_2$ fertilization (Wang et al., 2020) and the aforementioned LUC effect on NPP. This change provides a better match with known LUC effects on terrestrial biomass and production (Winkler et al., 2021; Malhi et al., 2004). More generally, it means that Hector does not regrow vegetation after LUC-driven deforestation; regrowth fluxes should be included in the LUC inputs (see above).

Fourth, Hector V3 also includes a novel implementation of permafrost thaw, a potentially significant process affecting the Earth system (Hugelius et al., 2020) that releases both $CO_2$ and $CH_4$ into the atmosphere. Hector's permafrost implementation was fully described by Woodard et al. (2021). Briefly, permafrost is treated as a separate land carbon pool that becomes available for decomposition into both $CH_4$ and $CO_2$ once thawed (Schädel et al., 2014). The thaw rate is controlled by biome-specific land surface temperature and calibrated to be consistent with both historical data and projections from the sixth phase of the Coupled Model Intercomparison Project (CMIP6) (Burke et al., 2020). Woodard et al. (2021) found that the fraction of thawed permafrost carbon available for decomposition was the most influential parameter in this approach and that adding permafrost thaw to Hector resulted in 0.2–0.25 °C of additional warming over the 21st century. The addition of permafrost to the V3 model produced changes in climate and permafrost carbon pools that are fully consistent with those reported by Woodard et al. (2021).

An optional new feature in Hector V3 is the ability to track the flow of carbon as it moves between the land and ocean carbon pools and the atmosphere (as $CO_2$). At a user-defined start date for tracking, the model tags all carbon in each of its pools as self-originating – e.g., the soil pool is deemed to be composed of 100 % soil-origin carbon. As the model then runs forward, the origin tag is retained as carbon is exchanged between the models' various pools; if 1 Pg C with origin $X$ is incorporated into a 19 Pg C pool with origin $Y$, for example, at the next time step, the 20 Pg C pool is tracked as 5 % origin $X$ and 95 % origin $Y$. At the end of a run, detailed information about the composition of each pool at each time point can be analyzed. This capability does not affect model behavior or any outputs, although it does impose a substantial performance penalty. Carbon tracking was described in detail by Pressburger et al. (2023) and is off by default.

### 2.2.3 Radiative forcing

At each time step, after Hector's carbon cycle solves for its fluxes and new pools and all GHG concentrations are computed, Hector calculates total radiative forcing as the sum of 39 forcing effects (listed in Table S1), each relative to the 1750 base year. The forcing effects of volcanoes and albedo are read in as inputs, along with a normally unused "miscellaneous forcing" input available for experimental manipulation. The remaining 36 forcing effects of various aerosols, aerosol–cloud interactions, pollutants, and greenhouse gases are calculated internally within Hector. The forcing effects of tropospheric $O_3$ and stratospheric $H_2O$ use the same calculations as Hartin et al. (2015). For the other forcing agents – $CO_2$; $CH_4$; $N_2O$; 26 halocarbons; aerosol–cloud interactions; and the effects of BC, OC, $SO_2$, and $NH_3$ – Hector V3 has adopted the forcing equations from the AR6 (see Table S5). Notably, the forcing effect from $NH_3$ was not previously included in Hector. In addition, the aerosol–cloud interaction forcing replaces the indirect effects of $SO_2$ forcing previously used to approximate the $SO_2$ and cloud interactions.

### 2.2.4 Temperature

In Hector V2, a 0-D energy balance model was replaced with a DOECLIM model (Vega-Westhoff et al., 2019). The DOECLIM model uses Hector's total radiative forcing to determine global temperature change. It is a four-box energy balance model, meaning that it models heat transfer within the climate system, which is represented by four idealized boxes: land (surface), air (2 m) over land, air (2 m) over the ocean, and sea surface (ocean mixed layer). The DOECLIM model uses a system of differential equations to model the temperature change in these four boxes in response to radiative forcing while accounting for the proportional differences in ocean and land masses and effective heat capacity (Tanaka et al., 2007).

In Hector V3, the DOECLIM model is a fully integrated component of the model, and its outputs now affect Hector's land carbon cycle: the DOECLIM model's land temperature drives heterotrophic respiration, while sea surface

temperature affects ocean carbon cycle dynamics. The difference between land and ocean temperature change, known as the land–ocean warming ratio, is an emergent property of the DOECLIM model and is used by default. Two additional parameters can be used to adjust the contributions of aerosols (BC, OC, $SO_2$, $NH_3$, and aerosol–cloud interactions) and volcanic forcing to global temperature. By default these are set to a value of 1, with the assumption being that the forcing–temperature relationship is consistent for all forcers. These scalar terms enable users to adjust the temperature sensitivity to aerosol and volcanic forcing in uncertainty analyses or when using Hector to emulate ESMs that exhibit different sensitivities to aerosol and volcanic forcings (Dorheim et al., 2020).

### 2.2.5 Constraints

Hector can run in "constrained" mode, which allows users to overwrite a specified Hector variable and replace it with a prescribed time series. Values can be prescribed for atmospheric $CO_2$ and all other GHG concentrations (effectively resulting in a concentration-forced run rather than an emissions-forced run). In addition, global temperature, total radiative forcing, and net biome production (effectively turning off the model's terrestrial carbon cycle) can also be constrained. In constrained mode, user-provided values can be seamlessly overwritten and replaced with internally calculated ones and are thus subsequently used by the downstream Hector components. For example, a Hector run that uses the total radiative forcing constraint will use the user-prescribed values, rather than Hector's internally calculated total values, to calculate energy fluxes and temperature change (see Table 2 for more examples and details).

The ability to run in the constrained mode is a useful feature that has a number of applications. For example, Hector's concentration constraints enable concentration-forced experiments, e.g., 1 % $CO_2$ and abrupt-4xCO2 experiments (Eyring et al., 2016), to comply with the RCMIP protocol (Nicholls et al., 2020). In addition, constraints facilitate coupling Hector with other models: the net biome production (NBP) constraint can be used to transfer global NBP values from a regional terrestrial carbon cycle model to Hector, after which Hector's ocean carbon cycle and climate dynamics can be calculated. Finally, running Hector in constrained mode can help diagnose model behavior. For example, concentration constraints can be used following a new model development that results in an unexpected increase in global temperature. Running Hector with constrained $CO_2$ concentrations or with total radiative forcing (RF) will help developers attribute this novel behavior to changes to Hector's carbon cycle or climate dynamics.

### 2.2.6 Model parameterization

Hector V3's default parameterization is mostly inherited from previous versions of Hector (Hartin et al., 2015; Vega-Westhoff et al., 2019), with the exception of when robust updated estimates are available. In particular, the V3 model uses more recent estimates published for preindustrial NPP, $CO_2$, $CH_4$, and $N_2O$ concentrations, as well as estimates of the preindustrial carbon cycle to initialize its ocean carbon pools (Table 1). Initial preindustrial sea surface temperatures used by Hector's ocean component were updated from a CMIP5 multimodel mean to a CMIP6 multimodel mean. Output files, containing historical ocean surface temperature data, from 24 CMIP6-participating models (see Table S11) were processed to compute the area-weighted mean temperature globally at both high ($> 55°$) and low ($\leq 55°$) latitudes from 1850 to 1860 (Table 1).

To calibrate the final model, five additional Hector parameters were fit to comparison data using a Nelder–Mead optimization routine (Nelder and Mead, 1965) in a two-part protocol. First, the natural $N_2O$ and $CH_4$ emissions, which are assumed to be constant throughout the run, were calibrated to the median AR6 $N_2O$ and $CH_4$ radiative forcing (Smith et al., 2018). Second, three Hector parameters – the $CO_2$ fertilization factor $\beta$ (unitless), heterotrophic respiration temperature sensitivity $Q_{10}$ (unitless), and ocean heat diffusivity $\kappa$ ($cm^2 s^{-1}$) – were fit to historic $CO_2$ concentrations (Meinshausen et al., 2017) and GMST (Morice et al., 2021) observations from 1850 to 2021. The Meinshausen et al. (2017) records consist of data for a single year in 1750 as well as a complete time series from 1850 to 2014. We chose to use $CO_2$ and GMST data because they are observed data with long time series; conversely, other potential records, such as ocean and land sink estimates, come from either inversions or models (Friedlingstein et al., 2023). The optimization routine simultaneously minimized the average of the two variables' mean squared errors between Hector $CO_2$ concentrations and GMST and these observed data. Parameter bounds (beyond which the optimizer was not allowed) were set at $\pm 2\sigma$ – i.e., for a normally distributed variable, $\sim 95$ % of the possible distribution was used. The best fits for $\beta$, $Q_{10}$, and $\kappa$ (Table 1) were then set as Hector V3's default parameters. The materials and scripts used to calibrate Hector are available in the repository for the paper (https://github.com/JGCRI/Dorheim_etal_2024_GMD, last access: 29 May 2024) to ensure the reproducibility and transparency of the calibration process.

## 2.3 Model runs and analysis

To assess model performance, we compared Hector results with both observations and ESM projections. For the historical period, we ran Hector in its default emission-driven mode, with inputs according to the RCMIP protocol (Nicholls et al., 2021, 2020) and the default parameteriza-

**Table 2.** Descriptions and summaries of the Hector constraints. The constraint name column shows the names as they appear in the model's INI (initialization) files.

| Name | Description | Implementation |
|---|---|---|
| CO2_constrain | Time series of $CO_2$ concentration values (ppmv $CO_2$) | $CO_2$ radiative forcing (RF) is calculated from the user-provided $CO_2$ concentrations and then used to calculate total RF and temperature. If needed, $CO_2$ is debited from or credited to the deep ocean to meet the $CO_2$ concentration constraint and satisfy Hector's global carbon cycle mass balance check. |
| CH4_constrain | Time series of $CH_4$ concentration values (ppbv $CH_4$) | $CH_4$ RF is calculated from the user-provided $CH_4$ concentrations and is fed into total RF and temperature. |
| N2O_constrain | Time series of $N_2O$ concentration values (ppbv $N_2O$) | $N_2O$ RF is calculated from the user-provided $N_2O$ concentrations. |
| $X$_constrain ($X$ is the identifier for 1 of 26 halocarbons modeled by Hector) | Time series of a single halocarbon concentration (pptv) | RF for the halocarbon $X$ is calculated from the user-provided concentrations. |
| RF_tot_constrain | Time series of total radiative forcing values (W m$^{-2}$) | User-provided total-RF values are used to calculate temperature and heat flux. In this case, the emission inputs do not drive model behavior. |
| NBP_constrain | Time series of net biome production (NBP) values (Pg C yr$^{-1}$) | User-provided NBP values are used to upscale or downscale NPP and heterotrophic respiration (RH) so that their total matches the constraint. This effectively bypasses the model's terrestrial carbon cycle. |
| tas_constrain | Time series of global mean air temperature values (°C) | User-provided temperature values are overwritten and replaced with Hector's, with a smooth transition between the constrained and free-running behaviors. |

tion described in the previous section. Hector's GMST results from 1850 to 2021 were compared with the HadCRUT5 (Morice et al., 2021) GMST observations, while Hector's $CO_2$ concentrations for the year 1750 and from 1850 to 2014 were compared with the CMIP6 (Meinshausen et al., 2017) $CO_2$ concentrations. We used the root mean square error (RMSE) to quantify the differences between model results and the observations. An ordinary least squares linear regression was fit to the Hector results and the observational data products to provide additional insights into the goodness of fit. An $R^2$ value close to 1 suggests a high degree of correlation between the Hector results and the observations.

For the future period, we first compared Hector's temperature with the AR6 near-term (2021–2024), mid-term (2041–2060), and long-term (2081–2100) warming projections. For this, Hector was run in emission-driven mode using the emissions from the RCMIP (Nicholls et al., 2020) protocol. Hector's near-term, mid-term, and long-term warming projections were computed as the 20-year averages using the model's GMST output.

Second, the model was run in constrained mode, in which concentrations for $CO_2$, $CH_4$, $N_2O$, and 26 halocarbons from the RCMIP (Nicholls et al., 2020) were prescribed and com-

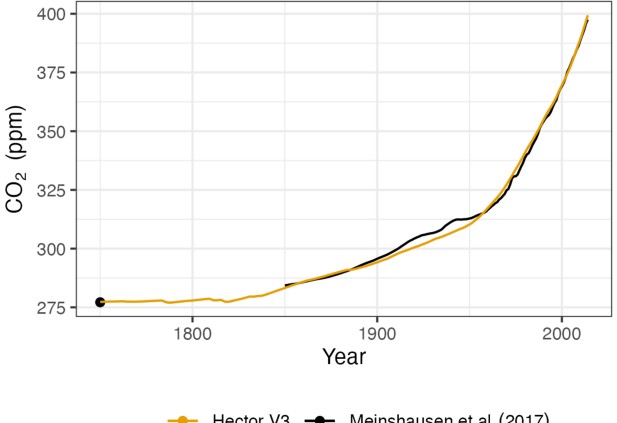

**Figure 2.** Hector's $CO_2$ concentrations (orange) compared with the observational product of the CMIP6 $CO_2$ concentrations (black; Meinshausen et al., 2017).

pared with CMIP6. These concentration-driven runs were consistent with the CMIP6 protocol (Eyring et al., 2016), allowing for a direct comparison of Hector's climate dynamics

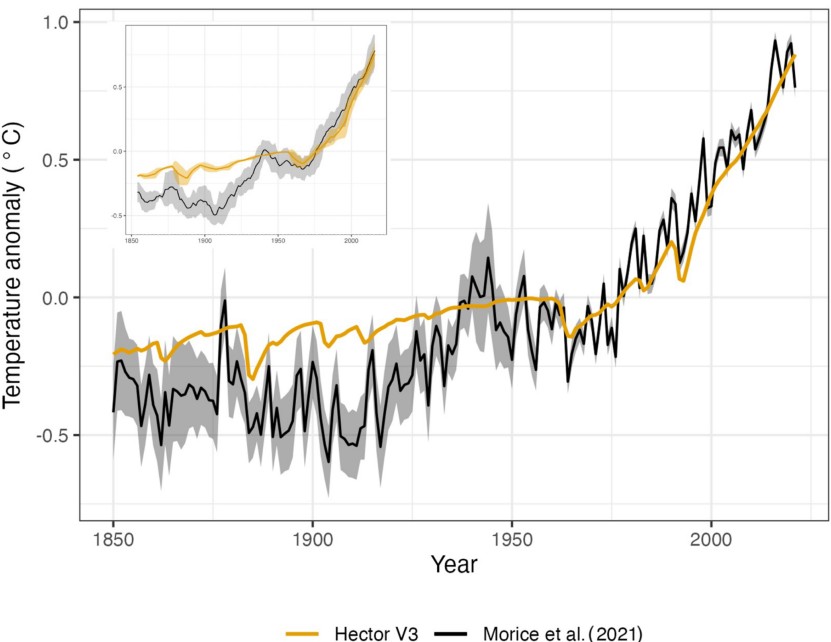

**Figure 3.** GMST anomaly relative to 1951–1980 for Hector (shown in orange) and HadCRUT5 GMST observations (Morice et al., 2021) (shown in black with associated uncertainty). The inset figure shows the rolling decadal average.

with those of the ESMs. For this step, output files from 15 ESMs were processed to compute area-weighted global air, land air, and sea surface temperature anomalies. The CMIP6 models were selected based on data availability for the variables and scenarios; a complete list of models is given in Table S12. We used the first available ensemble member since the internal variability between members was unlikely to affect long-term dynamics, which are the focus of RCMs (Eyring et al., 2016).

## 3   Results and discussion

Historical $CO_2$ concentrations from an emission-driven Hector run are compared with the Meinshausen et al. (2017) dataset in Fig. 2. The Hector results closely follow the observed values with an RMSE of 2.14 ppm $CO_2$ and a correlation coefficient of 0.99, indicating good agreement between Hector's output and historical carbon cycle observations. Figure 3 compares global mean temperature from an emission-driven Hector run with historical observations (Morice et al., 2021). The difference between Hector's results and the observations is an RMSE of 0.18 °C, which is less than the 0.36 °C standard deviation of the comparison dataset. The linear fit between the Hector results and the observations has an adjusted $R^2$ value of 0.87 (Fig. 3). The recent (2012–2021) decadal average of global mean surface temperature for Hector was $0.75 \pm 0.09$ °C. The model's most notable departure from the observational record is in the late 19th and early 20th centuries (Bauer et al., 2020;

Nicholls et al., 2020). The model also generally reproduces modern-day airborne fraction values (Jones et al., 2013; Pressburger et al., 2023). The model's modern (2014–2024) decadal averages of sea surface temperature and ocean pH are $0.78 \pm 0.08$ °C and $8.1 \pm 0.008$, respectively. Hector's land sink for 2013–2022 was $1.94 \pm 0.1 \, \mathrm{Pg} \, \mathrm{C} \, \mathrm{yr}^{-1}$, which is lower than the land sink of $2.9 \pm 0.9 \, \mathrm{Pg} \, \mathrm{C} \, \mathrm{yr}^{-1}$ reported by the Global Carbon Project (GCP; Friedlingstein et al., 2023) during the same decade. Hector's ocean sink of $3.08 \pm 0.13 \, \mathrm{Pg} \, \mathrm{C} \, \mathrm{yr}^{-1}$ is consistent with the GCP ocean sink of $2.8 \pm 0.4 \, \mathrm{Pg} \, \mathrm{C} \, \mathrm{yr}^{-1}$. Ultimately, we conclude that emission-driven Hector results are in agreement with historical temperature and $CO_2$ observations except, as noted above, for those from the latter half of the 19th century.

The comparison of Hector's historical results with observations is complemented by evaluating Hector's future temperature results against CMIP6-assessed (Fig. 4) and AR6-assessed warming projections (Canadell et al., 2021). For the future SSP1-2.6, SSP2-4.5, and SSP5-8.5 projections, Hector's temperature outputs fall squarely within the CMIP6 model spread (Fig. 4). In addition, Fig. 5 shows Hector's performance in two stylized experiments – 1 % $CO_2$ and $4 \times CO_2$ experiments relative to CMIP6 ESMs. These are baseline experiments from the Coupled Model Intercomparison Project (CMIP) Diagnostic, Evaluation, and Characterization of Klima (DECK) protocol (Eyring et al., 2016) that are designed to diagnose a model's climate sensitivity and feedback strength, provide an idealized benchmark for its transient behavior (for the 1 % $CO_2$ experiment), and characterize its climate sensitivity and fast-response performance

**Table 3.** Key emerging climate metrics, historical warming, effective radiative forcing, and future warming projections from Hector versus the IPCC AR6 "best estimates" from the AR6 (Table 7.SM.4). The Hector values were derived from runs using Hector's default parameterization in the free-running emission-driven mode for historical and SSP scenarios. The parenthetical IPCC AR6 values indicate the "very likely" (5 %–95 %) ranges of the AR6. Acronyms are given for equilibrium climate sensitivity (ECS), transient climate response to cumulative carbon emissions (TCRE), transient climate response (TCR), global surface air temperature (GSAT), well-mixed greenhouse gas (WMGHG), and effective radiative forcing (ERF) (Nijsse et al., 2020).

| Key metrics | | Hector | IPCC AR6 |
|---|---|---|---|
| ECS (°C) | | 3 | 3 (2, 5) |
| TCRE (°C per 1000 GtC) | | 1.51 | 1.65 (1, 2.3) |
| TCR (°C) | | 1.84 | 1.8 (1.2, 2.4) |
| Historical warming and effective radiative forcing | | | |
| GSAT warming (°C; 1995–2014 relative to 1850–1900) | | 0.73 | 0.85 (0.67, 0.98) |
| Ocean heat content change (ZJ; 1971–2018) | | 471 | 396 (329, 463) |
| Total aerosol ERF (W m$^{-2}$; 2005–2015 relative to 1750) | | −1.24 | −1.3 (−2, −0.6) |
| WMGHG ERF (W m$^{-2}$; 2019 relative to 1750) | | 3.87 | 3.32 (3.03, 3.61) |
| Methane ERF (W m$^{-2}$; 2019 relative to 1750) | | 0.54 | 0.54 (0.43, 0.65) |
| Future warming (GSAT; °C relative to 1995–2014) | | | |
| SSP1-1.19 | 2021–2040 | 0.73 | 0.61 (0.38, 0.85) |
| | 2041–2060 | 0.90 | 0.71 (0.4, 1.07) |
| | 2081–2100 | 0.72 | 0.56 (0.24, 0.96) |
| SSP1-2.6 | 2021–2040 | 0.75 | 0.63 (0.41, 0.89) |
| | 2041–2060 | 1.08 | 0.88 (0.54, 1.32) |
| | 2081–2100 | 1.10 | 0.90 (0.51, 1.48) |
| SSP2-4.5 | 2021–2040 | 0.75 | 0.66 (0.44, 0.90) |
| | 2041–2060 | 1.29 | 1.12 (0.78, 1.57) |
| | 2081–2100 | 1.98 | 1.81 (1.24, 2.59) |
| SSP3-7.0 | 2021–2040 | 0.76 | 0.67 (0.45, 0.92) |
| | 2041–2060 | 1.43 | 1.28 (0.92, 1.75) |
| | 2081–2100 | 2.94 | 2.76 (2.00, 3.75) |
| SSP5-8.5 | 2021-2040 | 0.88 | 0.76 (0.51, 1.04) |
| | 2041–2060 | 1.74 | 1.54 (1.08, 2.08) |
| | 2081–2100 | 3.79 | 3.50 (2.44, 4.82) |

(for the 4xCO2 experiment). Again, the model falls squarely within the CMIP6 model spread, with no suggestion of anomalous behavior. Hector's transient climate response to cumulative $CO_2$ emissions is 1.51 °C per 1000 Pg C, which is cooler than the IPCC-AR6-assessed best estimate of 1.65 °C per 1000 Pg C but falls within the "very likely" range of 1.0 to 2.3 °C per 1000 Pg C (Arias et al., 2021). In general, we conclude that the model exhibits climate responses consistent with the AR6 (Table 3).

## 4 Conclusions

In this paper, we documented the changes and new features of Hector V3. We showed that Hector's emission-driven historical results are generally consistent with observed $CO_2$ concentrations and global mean surface temperature, with the exception of late 19th-century and early 20th-century cool-

ing (Bauer et al., 2020). Hector's future projections of land, ocean, and global average temperature are consistent with a CMIP6 ensemble of models. Thus, we conclude that in the context of RCMs, Hector reproduces most global-scale historical trends and produces 21st-century projections that are consistent with Earth system models.

This fidelity to current climate observations and future CMIP6 projections means that there are many potential use cases for Hector, but it is important for users to understand both the advantages and disadvantages of using it compared to other RCMs or ESMs (Nicholls et al., 2021). The freely available R package and online interface facilitate its integration into both standard analytical pipelines and classroom settings, meaning that students can gain hands-on experience with running a climate model and interpreting results; such educational use is supported by the fact that Hector is a well-documented open-source climate model and that multiple

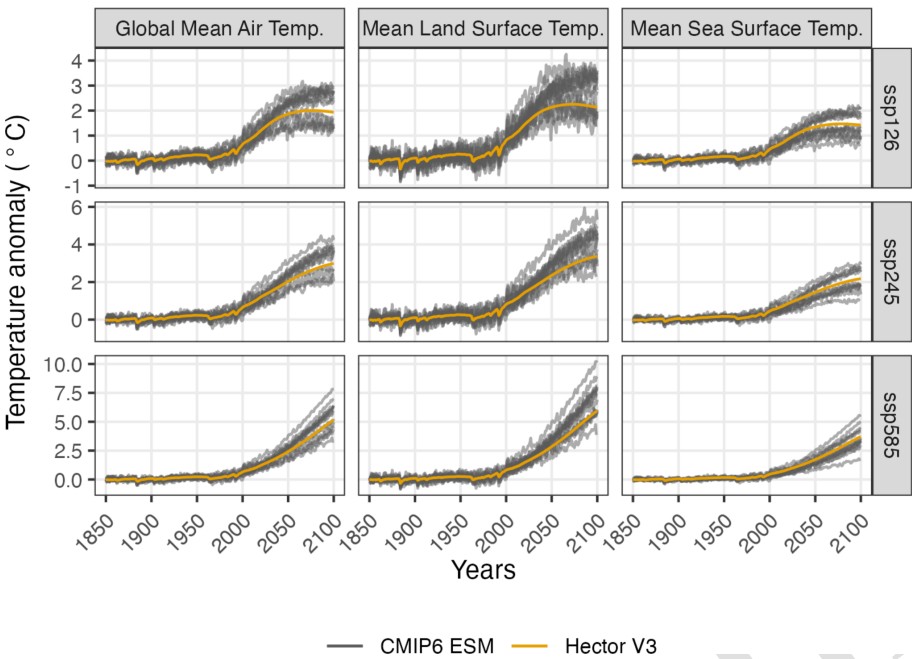

**Figure 4.** Global, land, and sea surface temperature anomalies relative to 1850–1900 from concentration-driven (constrained) Hector runs are shown in orange, while temperature output from 15 different CMIP6-participating ESMs is shown in gray (see Table S8).

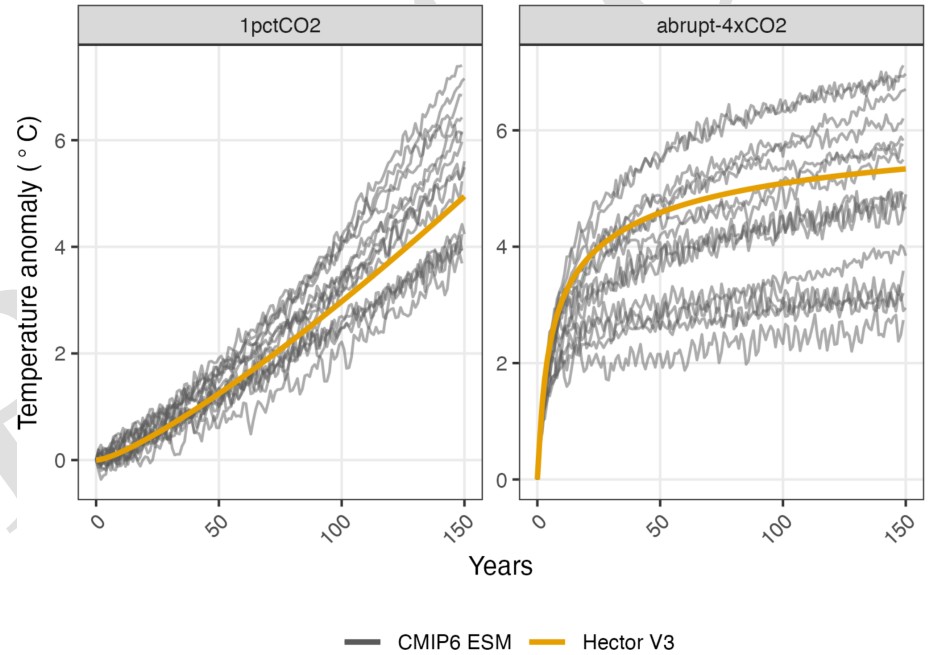

**Figure 5.** Global temperature anomalies from the stylized experiments (1 % $CO_2$ and 4xCO2) (Eyring et al., 2016) for Hector (orange lines) and 15 different CMIP6-participating ESMs (gray lines; see Table S8).

means of running the model are available (Hector UI, RHector, and C++ executable). The model's fully open-source C++ core is easy to couple with other models (Calvin et al., 2019). Using the Hector R package (https://github.com/jgcri/hector, last access: 29 May 2024), it is easy to generate and analyze large ensembles of Hector results, which can be used to explore uncertainty spaces (Nicholls et al., 2021; Pressburger et al., 2023). Finally, Hector's performance and open, flexible calibration procedure support efforts to emulate more complex ESMs in the facilitation of novel, com-

putationally intensive experiments (Lu and Ricciuto, 2019; Chen et al., 2023).

It is also important to note Hector's limitations. The model is more complex and thus harder to understand compared to approaches such as FAiR (Leach et al., 2021), although its complexity is comparable to that of MAGICC (Meinshausen et al., 2011). Hector does not account for the ocean's biological pump or changes in ocean stratification; whether these are compensating or compounding errors is unclear and warrants future research (Jin et al., 2020). Longer-term simulations are beyond Hector's scope, as is true for most RCMs, as the model's ocean does not include the heat storage changes that strongly affect long-term global temperature dynamics (Baggenstos et al., 2019; Abraham et al., 2013). Future work should aim to understand and rectify the differences between Hector's terrestrial carbon sink and other sources while remaining consistent with Hector's moderate complexity and goals; it will always be important to consider trade-offs between costs (i.e., increased complexity threatening interpretability, increased predictive uncertainty from additional model parameters, and computational efficiency) and benefits (increased fidelity and representativeness) (Sarofim et al., 2021).

Finally, in addition to continued science improvements, future versions of Hector will benefit from added infrastructure capabilities. The current parameter-calibration routine is relatively simple, and it may be worth exploring more sophisticated model-calibration procedures (Chen et al., 2023) in future versions of Hector. In addition, a turnkey ability to conduct probabilistic model forecasts (Fawcett et al., 2015; Ou et al., 2021), i.e., propagating parameter distributions and uncertainty (Pressburger et al., 2023) to produce probabilities of future climate change, is an important capability that a companion R package has been developed to handle (Brown et al., 2024). Leveraging this new capability for probabilistic projects will be important for future analyses using Hector to understand the changing Earth and climate system.

*Code and data availability.* Hector V3.2.0 was used to generate the Hector results analyzed and the figures included in the main text and in the Supplement. This version of Hector is available at https://github.com/JGCRI/hector (last access: 29 May 2024) via the V3.2.0 release and is archived at https://doi.org/10.5281/zenodo.10698028 (Dorheim et al., 2024); this includes all the initialization, emission, and concentration files. All of the code and data used to calibrate Hector, perform all model runs, and produce data visualizations are available at https://github.com/JGCRI/Dorheim_etal_2024_GMD (last access: 29 May 2024), and the "GMD3" release associated with this iteration of the paper is archived at https://doi.org/10.5281/zenodo.10698925 (Dorheim, 2024).

*Supplement.* The supplement related to this article is available online: https://doi.org/10.5194/gmd-17-1-2024-supplement. TS1

*Author contributions.* KD, BBL, SJS, SG, RG, CH, LP, ANS, and DW all contributed to Hector's development. CT and SJS helped conceptualize the model experiments. KD and BBL led the preparation of the original draft, and all coauthors contributed to the final draft.

*Competing interests.* The contact author has declared that none of the authors has any competing interests.

*Disclaimer.* The views expressed in this article are those of the authors and do not necessarily represent the views or policies of the U.S. Department of Energy, Environmental Protection Agency, or National Aeronautics and Space Administration.

Publisher's note: Copernicus Publications remains neutral with regard to jurisdictional claims made in the text, published maps, institutional affiliations, or any other geographical representation in this paper. While Copernicus Publications makes every effort to include appropriate place names, the final responsibility lies with the authors.

*Acknowledgements.* This research was supported by the U.S. Department of Energy's Office of Science as part of the MultiSector Dynamics, Earth and Environmental System Modeling Program. The authors would also like to acknowledge the EPA project (grant no. DW-089-92459801-8 TS2) for contributing to the radiative forcing updates included in Hector v3. The authors would also like to acknowledge Robert Link and Sven Willner for their contributions to Hector and their work on RHector and pyhector, respectively.

*Financial support.* This research has been supported by the U.S. Department of Energy as part of the MultiSector Dynamics, Earth and Environmental System Modeling Program. The EPA project (grant no. DW-089-92459801-8 TS8) supported the radiative-forcing updates TS3.

*Review statement.* This paper was edited by Marko Scholze and reviewed by Benjamin Sanderson and three anonymous referees.

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
