# Peer review of "Hector V3.2.0: functionality and performance of a reduced-complexity climate model"

_EGUsphere, 2023_

## Author Response (AR1)

23 February 2024

Thanks for your consideration of our submission egusphere-2023-1477, "Hector V3.1.1: functionality and performance of a reduced-complexity climate model". We are grateful for the four reviewers' many thoughtful questions and suggestions. Our responses below are in **bold**; line numbers refer to the final (clean) version of the revised manuscript.

This revised version has been extensively revised with more detail on the model's governing equations, calibration procedures, and performance—including in stylized DECK-style experiments, as suggested by several reviewers. Clarity has been improved throughout, citations updated and checked, and the figures improved.

We hope that the revised manuscript addresses all concerns, but of course welcome further feedback.

Thank you,
Kalyn Dorheim, for the authors
* * *
**Referee 1**

It is undoubtedly good to have an updated description of Hector, although the execution could be much better.

I'm unsure where this paper stands in the editorial policy of GMD, because it is neither a model description (not detailed enough) nor a model validation (not thorough enough). The simulations look more illustrative than anything else, and as a reader it's hard to draw any conclusion that goes beyond "okay, Hector gives results that are not absurd compared to IPCC".
I do not know where the bar for GMD is, exactly, but I would ask for additional analysis and better organization of the paper before accepting publication. A clearer assessment of strengths and weaknesses would help position the model against its "competition". Even if the goal is to simply demonstrate the adequacy of the model, I feel some discussions and elements are missing to conclude on this without a doubt.

Disclaimer: I did not review any of the code provided with the paper.

**We appreciate the overall positive sentiment, and have worked to provide more detail on the model's parameterization, functionality, and performance; restructure the manuscript for clarity; and discuss Hector's strengths and weaknesses in more depth. Please see specific responses below on all of these points.**

Specific comments:

Section 2.1. I expect such a description to be exhaustive, and to avoid using example lists (marked by the use of "e.g.", "such as", and so on). Examples of this are on l.69 where all short-lived should be listed, or on l.75 where it is unclear whether the two forcings are the only ones.

**We agree. The text has been revised throughout to eliminate any ambiguity, and we have also checked for and fixed similar problems elsewhere.**

I also feel some equations from Hartin et al. (2015) could be repeated here, as it is extremely annoying for the reader to have to read this paper along with the previous one. For instance, on l.65-66.

**This issue was raised by other reviewers as well. We now include (mostly as supplementary material) all model equations, and reference these in the text; see lines 61, 64, 66, 70, 174, and 180.**

Section 2.2. I would split this between subsections that deal with the physics and its formulation, and subsections that deal with technical and implementation aspects.

**We appreciate the suggestion, but on balance think that the current subsection organization—covering software, carbon cycle, radiative forcing, temperature, constraints, and parameterization—is reasonable.**

Section 2.2.2. More details on the carbonate variables is required. What system of equations is used? what solver? is it based on a third-party tool? is it an emulation in itself? The answers might be in previous papers, but it doesn't preclude giving information here.

**We have significantly expanded the description of the ocean model, including its carbonate system, and added equations and initialization values. In some cases we still fall back on referring readers to previous publications—e.g., for the equation details of the ocean's surface chemistry or how the permafrost thaw was parameterized—but we hope that we have struck a reasonable, and better, balance in this area.**

"Hector can run with an arbitrary number of independent biomes." Okay, but is it an option in the model as it is available? i.e. are alternative parameterizations with several biomes readily available? Or is it just a structural option?

**We have clarified this (lines 122-124).**

LUC:
The way LUC is actually treated in the model is very obscure and requires a lot more discussion. The manuscript mentions that two time series are provided (gross positive and gross negative). But are these two treated differently? Is there any difference between inputting two gross fluxes or one net flux?

**Thanks for catching this. Separating the fluxes allows users more flexibility, but also, yes, there's a difference: losses are removed from vegetation, detritus, and soil, but uptake fluxes are added only to vegetation. This has been clarified in the text (126-142).**

What is the source of these two fluxes? One key issue I see, here, is that gross positive and negative fluxes of LUC are often linked together and cannot be set independently of one another. For instance, C emissions from harvested wood products' oxidation have the same origin as the biospheric regrowth triggerred by the harvest in the first place. It's essentially impossible to change one without changing the other.

**It is true that sometimes these fluxes are linked, but sometimes they are not. The most obvious case is perhaps afforestation, which from a high-level point of view adds to the vegetation pool and thus NPP; some of that NPP will subsequently become respired, of course, but in prior versions of the model users had no way to look at afforestation versus deforestation effects.**

The second point (on l.127) is absolutely unclear. It should be expanded.

**We have expanded it (137-142). The point is that the carbon-cycle effects are not well constrained (Yue et al. 2020), and that for a non-spatial model such as Hector the pool sizes are governed by the total amount of carbon in the system and the first-order equations linking the pools; LUC loss is only temporary until the pools re-equilibrate. So it seemed simpler to adopt a proportional loss approach and remove three parameters that did not, ultimately, do much.**

The third point (on l.129) doesn't seem scientifically validated. The modelling choice is, if I understand it correctly, that NPP is reduced by a factor equal to the ratio of cumulative LUC emissions over preindustrial land C pool. What is the justification of this?

**The intended justification is that deforesting land use changes photosynthesis and thus NPP (Malhi et al. 2004; Kaplan et al. 2012). For example, if one deforests the Amazon, global NPP should be reduced, but in previous versions of Hector that didn't happen—NPP remained fixed.**

Has it been derived from complex models? How does it impact the land sink, precisely? Does this roughly reproduce the loss of sink capacity estimated in the global carbon budget (their section D4)? The Winkler et al. (2021) paper cited here doesn't bring any answer to these questions. Also, the ref is missing from the references.

**We apologize about and have fixed the missing Winkler reference. As the manuscript notes (now in expanded detail, lines 146-151), this is in accord with our understanding of how LUC affects available photosynthesis and also improves the model's LUC-driven performance.**

Carbon tracking:
I have seen several presentations of the carbon-tracking capability of Hector, and I remain somewhat doubtful of the implementation. Since I don't see anything in Pressburger et al. (2023) to alleviate my concerns, I will ask my questions here.
If one runs a piControl experiment with the tracking turned on, would one see pool compositions change?

**An important point that we wish to emphasize: carbon tracking does not change model outputs. This continues to be noted in the manuscript (170).**

If so, what is the meaning of this change, given that the C-cycle essentially stayed in steady-state? Also, is this steady-state change accounted for in transient tracking runs, e.g. by correcting the transient allocation with the control one?

**As noted above, the primary model outputs do not change with tracking off or on. What changes is that extra tracking information is written out in the latter case.**

I have the intuition that a change of structure of the model would change the allocation. Is it the case? For instance, the land C-cycle seems to start with NPP, which is a transfer from the atmospheric to one land pool. But what would happen if the model represented GPP explicitly? As NPP = GPP - Ra, the NPP would be the result of two opposite transfers that happen faster. Would the result of tracking be entirely different? If so, what is the meaning of it?

**This is an interesting question. Certainly if the structure of the model changes, that affects how carbon moves around the system, but the net tracking effect of the referee's GPP and NPP example was not obvious to us, and so we built a toy spreadsheet based on Hector's basic NPP/Rh system for land-atmosphere exchange to test how the dynamics might change. It turns out that no, there would be no change assuming that Ra draws from current-year GPP: the toy system converges to the same pool mixtures for both the current NPP-based system and a GPP/Ra implementation.**

**The spreadsheet is here should the referee want to examine it: [https://docs.google.com/spreadsheets/d/1j3jFRCfDAaLyf4kjrKrq7LFQU7HtxHgE/edit?usp=drive_link&ouid=115421369330155382886&rtpof=true&sd=true](https://docs.google.com/spreadsheets/d/1j3jFRCfDAaLyf4kjrKrq7LFQU7HtxHgE/edit?usp=drive_link&ouid=115421369330155382886&rtpof=true&sd=true)**

It is my understanding that the main interest is to track fossil C (and not exactly anthropogenic C, because of land use and other conceptual issues). Can the tracking be compared to and/or constrained with delta 13C estimates? Is there an actual link between the two, or is it conceptually too different?

**Tracking fossil C is one application, and the one highlighted in Pressburger et al. (2023). But we think the more compelling long-term application would be δ13C. The important**

**development needed will be to have processes such as photosynthesis discriminate against the heavier isotope (carbon-tracked pool). This is on our roadmap for Hector v4.**

Section 2.2.6. Why use 1850-1860 for this parameter (on l.216), which seems too short and with already changing background conditions, while piControl was available?

**We now use 1850-1900 as a baseline, per multiple reviewers' suggestions.**

Section 2.3. I think a better way to handle the CMIP6 higher-than-average ECSs would be to take the assessed temperature projections of the AR6 and the raw projections from CMIP6.

**Thank you for the suggestion; we're no longer comparing our results to the CMIP6 range based on ECSs. Figure 4 now shows CMIP6 output from individual models, and we compare Hector's future warming with the AR6 values in Table 3.**

Section 3. I seem to understand that scenarios were run in emission-driven mode, but CMIP6 models were in concentration-driven mode. This makes comparison harder. Does this also introduces biases? Resulting temperature could be right because of compensating biases in the C-cycle and climate modules!

**Our apologies that this was not clearer. In fact, all CMIP6 comparisons were made using Hector running in concentration-driven mode for (as the reviewer notes) consistency. We have clarified this in the text (259-265).**

In addition, this section needs a table with the main carbon and climate metrics, such as ECS, TCR, TCRE, betas and gammas, compared to AR6 and/or CMIP6. Therefore, I think additional simulations with idealized experiments such as 4xCO2 and 1%CO2 are warranted.

**Thank you for these two excellent suggestions. We have added a new summary Table 3 and replaced the previous Figure 5 with a new one showing the results of 4xCO2 and 1%CO2 simulations.**

Similarly, figures (and associated discussion) displaying the behavior of the C cycle are needed. The historical time series are not very informative, and additional historical C cycle comparison would help. But more importantly, projections of the C cycle vs. CMIP6 models should be discussed. Running the model in c-driven mode would help analyse carbon and climate systems separately.

**The model was run in concentration-driven mode, as noted above, and this has been clarified. We now include both a tabular summary (Table 3) and the results of running Hector in stylized DECK experiments that characterize the model's fast-response behavior, feedbacks, and sensitivity (Figure 5).**

Section 4. I disagree with some of the conclusion: this paper does not show the emulation capabilities of Hector. It just shows that projections with Hector are in the same ball park as CMIP6 models. This can make it a useful model, yes. But the model's performance in this regard should not be overstated.

**This is a good point, and we have toned down this language (295-300).**

I would also appreciate to see mention of the lack of a probabilistic setup upfront, rather than only in the conclusion.

**Thanks for the suggestion, but given that the goal of this work is to document the latest version of Hector (lines 16, 44-47), we think this section is the appropriate place to discuss the model's current limitations.**

Figure 1. I don't mean to have the authors rename variables in their code, but "geological" would sound much better than "Earth" pool.

**Agreed, but we fear that renaming it now would be confusing. We have added a clarifying note about this (line 100).**

Table 2. A bit unclear why some parameters would be displayed here and others not. How many parameters in total? Also, how do the carbon pools change after spinup?

**We have clarified (in the table caption) why only some parameters are included in Table 2. We have also added a note early in the methods about how carbon pools change during spinup; per the referee's question, this is how much they change after a default V3 spinup:**

| Component | Variables | inital_pool | post_spinup | Diff | Diff (%) |
|---|---|---|---|---|---|
| Land | veg_c | 550 | 562 | -12 | -2% |
| | detritus_c | 55 | 62.3489 | -7.3489 | -13% |
| | soil_c | 1782 | 2030.59 | -248.59 | -14% |
| | permafrost_c | 865 | 865 | 0 | 0% |
| Atmosphere | CO2_concentration | 277.15 | 277.15 | 0 | 0% |
| Ocean | preind_interdeep_c | 37100 | 35902.36 | 1197.64 | 3% |
| | preind_surface_c | 900 | 964.741 | -64.741 | -7% |

Figure 2. With so thick a line, the data seem to match even better than it probably does. Also, AR6 provides a better guess of historical $CO_2$ atmospheric concentration than Meinshausen et

al. (2017). Some uncertainty ranges somewhere on such a figure would make it look more scientific.

**We have modified this figure's line thicknesses, and as noted above simplified the model calibration procedure. The Meinshausen (2017) dataset remains our primary point of reference because of its role in CMIP6. There's not really an uncertain band to add for the historical data—we tried using differences between AR6 and Meinshausen to plot uncertainty, but they're numerically tiny. It is worth noting that it is quite common in this context to plot historical model performance without uncertainties, for exactly this reason; see for example figures in the relevant papers for MAGICC (Meinshausen et al. 2011) and FAIR (Smith et al. 2018).**

Figure 3. Same about uncertainty, maybe decadal estimates would be nice, since Hector doesn't have internannual variability.

**We have added an inset plot showing decadal estimates, which we agree is helpful.**

Figure 4. Here, in ssp126, one can clearly see a difference of trend by the end of 2100. I'd be curious to see a ssp524-over figure, as I suspect the model's dynamic is strongly dictated by its structure. This probably warrants a bit of discussion: while the numbers are in right ball park over the 21st century, the trend doesn't seem so great. Similarly, extension to 2300 would bring interesting insights.

**Thank you. This comment helped us identify a problem in the model's parameter-calibration step. The improved calibration procedure is described in the methods section (lines 228-242) and the model now exhibits much improved end-of-century behavior (Figure 4) for ssp126.**

Figure 5. I find it mostly uninformative. Being able to reproduce such a trivial emergent property of the system doesn't say much about the performance of the model. While the figure could stay, as it is indeed a result of the model, more detailled results seem warranted, for instance under idealized experiments like 4xCO2 and 1%CO2.

**We agree, and have replaced this figure with one showing the results of the idealized experiments suggested by the reviewer.**

Supplement Section 1. This is essentially a copy of the AR6 chapter 7 SM. That's not an issue per se, but it's surprising because of all the efforts the authors put into not writing any of Hector's equations. I'd rather have a summary of all equations in this Supplement.

**The supplement now provides a detailed summary of Hector's governing equations.**

Supplement Table 9. This is worth showing in the main text.

**We agree, and it has become the new summary Table 3.**

Technical corrections:
l.13. First sentence doesn't have a verb.
l.14. What is a "source of climate information"? It's a model. Period.

**Thanks, and agreed. These first two sentences have been fixed and improved (lines 14-16).**

l.98. I do not know what Boost is and what it does. More info needed.

**We concluded that this reference was unnecessary, and have removed it.**

l.106. "E.g."

**This was removed as part of addressing the comment about the carbonate process above.**

l.146. Why is permafrost turned off by default? Aren't the authors confident about that module?

**At the time this was written, the permafrost implementation had not been fully tested for its effect on the default model's parameterization and performance; leaving such 'beta' features off by default, but still available if desired, is a common practice in both geoscientific modeling and computer software more generally. Hector's permafrost component has now been fully tested and is on by default, and we have thus removed this sentence.**

l.163. Does that mean BC on snow is zero by default? Strange.

**Thanks for this catch, which was the product of poor wording. No, Hector has no black carbon on snow input, but it does have an "other RF" input that can be used to represent ancillary effects on radiative forcing. We have removed the reference to BC on snow, and clarified the "miscellaneous forcers" sentence (lines 72 and 175).**

l.165. Why were O3 and stratospheric H2O not updated on AR6, as all the rest obviously was?

**Because it wasn't possible to do everything in a single development cycle, unfortunately; inevitable time and budget limitations meant that things had to be prioritized, and a small number of tasks had to be deferred.**

l.177. "Improved" versus what?

**We have removed this language.**

l.183. "and is used by default". Would delete or rephrase.
l.185. ", and,". Delete.
l.188. "forces" +r.

**These have all been corrected.**

l.191. I'd rather call these "prescribed" variables than "constrained" ones, but that may just be me.

**If we could go back in time, we might also choose that terminology; but we feel that changing this now risks confusion. We have clarified this in the paragraph (lines 200-204).**

l.252. 15 ESMs is actually not that many.

**We appreciate the opinion.**

l.263-276. Repeated paragraph. I literally hate to see this. It just shows how little care was put into the paper…

**This mistake has been corrected.**

l.498. Reference period for temperature is typically 1850-1900.

**We have adopted a 1850-1900 baseline for consistency with other studies.**

Supp. l.17. "SSARF" has one too many S.
Supp. l.53. Units are wrong: "C", "S" and "NH3" should be after "Tg".

**Thanks for catching these mistakes, which have been corrected.**
* * *
**Referee 2**

This paper provides a brief introduction to v3 of the Hector simple climate model and displays some top-level results.

I found the paper a little incomplete. Generally, for model description papers for SCMs, a reader should theoretically be able to code up the model from equations provided or be able to easily tell what differs from other models. For example:

In section 2.1, the reader is pointed towards Hartin et al. (2015) – would it be too much duplication to transfer these equations (line 66) into this section? If it is, could you at least explain in general terms what Hector uses – a simple one-box decay model? The supplementary table 2 seems to suggest so, which is provided without context.

**All referees mentioned this point, and we have greatly expanded the amount of information provided: there is more process detail in the methods, and greatly expanded supplementary material, including most equations governing Hector's behavior. In one case, the ocean's carbonate chemistry, we still fall back on referring readers to previous publications (Hartin et al. 2016)—for the equations details of the ocean's surface chemistry or how the permafrost thaw was parameterized—but we hope that we have struck a reasonable, and better, balance in this area.**

If we assume that most GHGs are treated as a one-box decay model, the interesting structural differences between Hector and other SCMs will occur in the carbon cycle, methane cycle, and assumptions made for ozone and aerosol forcing. The carbon cycle is covered in this paper but burying the aerosol forcing equations in the supplementary material (manuscript, line 70), and not describing the ozone (line 165) and methane lifetime relationships, is quite unhelpful, since this is fundamental to how the model behaves and likely to be one of the key differences relative to other SCMs. Furthermore, the default methane lifetime and the sensitivities to emissions/concentrations of other species for both methane lifetime and ozone forcing should be documented somewhere, and whether these are fixed or tunable parameters.

**These details are now provided in the greatly expanded supplementary material.**

On this subject, there is not much detail about how flexible and tunable Hector is compared to other SCMs, and whether you could produce a probabilistic ensemble of parameter sets that span the range of ECS simulated by CMIP6 models or assessed by the IPCC – therefore sampling climate uncertainty - while still recreating observed warming and CO2 concentrations.

**We have added more information on these points, referencing both in-press work on a probabilistic software package for Hector (Brown et al. 2024), as well as earlier probabilistic work (Pressburger et al. 2023).**

Pressburger et al. (2023) kind of do this I suppose (I did not review in detail the reliability of the projections made in that paper), but showing just one ensemble member that, if I'm being slightly unkind, doesn't do a great job of recreating past warming (why have the authors used a 1951-1980 baseline in figure 3, when the benchmark proxy pre-industrial convention is 1850-1900? Hector would clearly underestimate historical warming under the earlier baseline), doesn't give me the confidence that this calibration of Hector is one that I would want to use to make reliable statements about future climate change.

**The choice of baseline is important for interpretation but, as long as the model and data source are using the *same* baseline, won't affect assessment of model performance. We**

**agree that Hector has some weaknesses recreating 1850-1950 warming patterns, similar to many other climate models, and now note this in the text (274 and 297-300). More generally, we note the model's limitations (314-319). We have adopted a 1850-1900 baseline for consistency with other studies. Finally, Figure 4, and the new Figure 5, are intended to give confidence that the model is highly consistent with CMIP6 across a wide range of emissions scenarios.**

Section 2.2.2: Why are results including permafrost not shown? It's kind of a shame to introduce a new feature and not showcase it. A with versus without plot of global mean surface temperature would be fine.

**At the time this was written, the permafrost implementation had not been fully tested for its effect on the default model's parameterization and performance, but that is no longer the case. Hector's permafrost component has now been fully tested and is on by default, and we have thus removed this sentence.**

Section 2.2.4: Does the DOECLIM model simulate the warming of the deep ocean? From which it would be able to calculate ocean heat content, TOA energy imbalance and thermosteric sea level rise.

**DOECLIM calculates ocean heat uptake but Hector currently does not have a deep ocean heat storage capability, and heat flow is only one way (from atmosphere into ocean0. This would be an obvious and severe problem for long-term simulations, but such millennial runs are generally outside the ambit of simple climate models. We have added a note about this to the concluding section (lines 317-319).**

Section 2.2.6: For CO2, some of the historical record was held back for "validation". I'm not sure I understand the logic of this. It rightly wasn't done for global mean surface temperature.

**We have simplified and clarified both this language and the calibration procedure itself (228-242), making the temperature and CO2 procedures consistent in not holding back any data (although still comparing with other time series not used for calibration). See also Figures 2 and 3.**

The last paragraph in this section is duplicated.

**We apologize for the oversight, and have fixed this.**

Section 3: A summary results table is necessary for ECS, TCR, TCRE (all emergent parameters?), present-day warming, present-day aerosol forcing, airborne fraction, and a few other key metrics. The SCM benchmark table of IPCC Chapter 7 SM would be a good template, or a 4-column table of variable, Hector result, target result, source of target. Example: Global mean warming 2012-2021 relative to 1850-1900 | 0.99 ± 0.1 °C | 1.0 °C | RCMIP phase 1 (Nicholls et al. 2020). [For this result in particular, I'm curious to where the uncertainty of ± 0.1°C

came from as in this paper Hector was not obviously run in probabilistic ensemble mode.] The summary results was half-done in the supplement with future warming.

**We agree, and now provide a new Table 3 with summaries of key metrics.**

It should be noted that the IPCC future warming projections were performed with concentration-driven runs and I assume Hector did these runs emissions-driven, so they may not be perfectly comparable.

**Our apologies that this was not totally clear. In fact, all CMIP6 comparisons were made using Hector running in concentration-driven mode for (as the reviewer notes) consistency. We have clarified this in the text (254-265).**

Somewhere, a comment should be made on model run time. How long does it take to run one (1750-2100) simulation? What about 1000, or 1 million? Does it parallelize well?

**Thanks for the good suggestion. We have added several sentences describing the model's performance (lines 95-96).**

All in all, I don't come away from this paper feeling convinced that Hector is any better than existing tools in the literature for climate assessments right now. That might be OK, but if so I also didn't get what Hector's USP is. Why should I drop the other models and use it? What can it do that MAGICC, FaIR, OSCAR etc. cannot?

**In lines 302-312 we now try to emphasize what we see as Hector's primary strengths and reasons for use: it's fully free and open source software; it has enough complexity and flexibility to run unusual experiments such as DACCS; it has unusual features in its terrestrial (e.g., permafrost) and ocean (e.g., surface chemistry) submodels; and its availability as an R package and interactive website greatly broadens the potential audience.**

Minor points:

Line 13: the first sentence is a bit odd.

**Thanks for catching this; fixed.**

Line 62: I'd like to see these listed or referred to. I assume this is what is in table S1.

**We now provide a reference (line 64).**

Line 109: what is this carbon value?

**These carbon values are given in Table 1, and we have added a reference to it in line 116.**

Line 122: extra close bracket

**Removed.**

Line 128-129: "provides a smoother, more intuitive model behavior…" Non-users of Hector have no context to compare this to. A plot showing the new and old treatments would be useful.

**This was confusing, and has been removed.**

Line 185 and 187: blank spaces where there should be, presumably Greek, variable names
Line 188: forcers

**These have been fixed.**

Line 307: Cumulative emissions of long-lived greenhouse gases…

**With the removal of the old Figure 5, this sentence has been removed.**

Line 335: compared to …?

**This is no longer applicable, given changes in the model's performance with the inclusion of permafrost and parameter-calibration routines, and has been removed.**

Units in table 2: Since C is used for carbon, use either °C or K for temperature.

**Good point; we have adopted °C throughout.**

Figure 2: lines are very thick!

**Agreed. We have modified this figure's line thicknesses, clarified the caption, and now include an uncertainty band around the Meinshausen (2017) line.**

Figure 4: SSP1-2.6 is quite overshooty, which seems to differ from ESM behavior. This is one of those plots that makes me look at Hector's warming profile and think that something isn't quite right here.

**This comment helped us—thank you—realize there was a problem with the model's parameter-calibration step. The corrected calibration procedure is described in the methods section (lines 228-242) and the model now exhibits much improved end-of-century behavior (Figure 4) for ssp126.**

In a lot of places the references are (author, year) where they should be author (year).

**Thanks; we have carefully checked all citation formatting.**

Supplement: the values used for rho_aci, s_SO2 and s_{BC+OC} need documenting, and how these values were obtained. They are very important in defining how aerosol forcing behaves historically and into the future, and balancing ECS/historical warming.

**The supplement has been greatly expanded with documentation about these and other points.**

Supplement eq 5, 7, 9, 11: this is effective radiative forcing, the delta term is the radiative adjustment as a proportion of SARF.

**Thank you. In general Hector radiative forcing is effective radiative forcing.**
* * *
**Referee 3**

Dorheim et al details v3 of the HECTOR simple climate model, which provides a number of modifications on previous versions in terms of both software interface and science.  As other reviewers have noted, the paper is largely descriptive, and requires cross-referencing with previous version descriptions and sub-component studies in order to fully describe the model logic.

**We have worked to provide more detail on the model's parameterization, functionality, and performance; restructure the manuscript for clarity; and discuss Hector's strengths and weaknesses in more depth. Please see specific responses below on all of these points.**

Perhaps more critically, the paper lacks some specific details on the changes which have been implemented in this version - on land use, radiative forcing from ammonia and aerosol cloud interactions.  In revision, the authors should take care to ensure that these new processes are fully described in sufficient detail to understand the new parameterisations.  Each science development should probably have its own subsection (only radiative forcing does at present), and a section on the comparative performance of the previous and revised version of the model would be useful.

**This issue was raised by other reviewers as well. We now include (mostly as supplementary material) all model equations, and reference these in the text.**

The land-use parameterisation sounds quite promising, but more detail is needed.  What are the input units for land use changes?   Is coupling with the natural carbon cycle only manifested through the NPP modifier in Eq1?  How do land use transitions interact with the natural carbon pools in the mode?

Can NPP be arbitrarily inflated to unphysical values given sufficiently large afforestation fluxes?

**Yes, it could be. But this is true of any input—give the model bad enough data, and it will produce bad results.**

The model calibration section could also do with sharpening.  It is not clear how global mean temperatures used to assess the model are independent of those used to calibrate the model, and the seperation between pre- and post-1959 concentrations seems a little arbitrary.  The periods are not independent in a strict sense, given they are part of a continuous smooth timeseries - but it's also not clear why we should care if they are independent, given that overfitting isn't really a risk here given the number of degrees of freedom involved.  Also - why is different logic used for the time partitioning of temperature and concentration data?

**We have simplified and clarified both this language and the calibration procedure itself, making the temperature and CO2 procedures consistent in not holding back any data (although still comparing with other time series not used for calibration), and fixed a mistake in the process. See lines 228-242 and Figures 2 and 3.**

The model seems to underestimate warming between the mid 19th and 20th centuries - which should be discussed a little more.  The decisions to only use CO2 and GMTS in calibration should also be discussed.  The authors have access to ocean and land sink data, for example - why not use it?

**Hector does run too warm (we assume this is what was meant?) in the mid 19$^{th}$ to early 20$^{th}$ century. This is now discussed more in the results and conclusion. We chose to use CO2 and GMST for calibration because they are observed data with long time series; conversely, ocean and land sink estimates come from either inversions or models (Friedlingstein et al. 2023). We now note this in the text (lines 267-281 and 296-297).**

The future experiments also seem to suggest that the default model behaves unlike most ESMs in high mitigation scenarios - with temperature reductions after 2060.  It's hard to make out in Figure 3 - but a quick glance at CMIP6 SSP126 ensemble members suggests only one model (MRI) exhibits this kind of cooling before 2100 (see below).

It would be good to at least understand this behaviour, and whether it's a tuning or structural issue - but at the very least it should be noted and flagged for future study.  Given these are c-driven runs, this must be an issue with the thermal dyanamics of the model - likely necessitating a wider parameter exploration of the ocean model heat transfer parameters and perhaps an updated calibration strategy.

**This comment helped us—thank you—realize there was a problem with the model's parameter-calibration step. The fixed calibration procedure is described in the methods**

**(282-242) and the model now exhibits much better end-of-century behavior (Figure 4) for ssp126.**

The authors should avoid the use of 'not very likely' as a descriptor of ESMs with climate sensitivity outside of the IPCC AR6 range. Firstly, this is not formal IPCC language and might lead to confusion. Secondly, the likelihood of a model's ECS does not translate directly to the likelihood of its outcome under a given scenario. More specifically, TCR is generally considered a more relevant metric for future warming in 21stC concentration driven scenarios (see Nijsse et al 2020, and classically Frame et al 2006 - though Grose et al 2018 empircally found the opposite, but didn't explain their result).

**We have removed this terminology in the revised ms.**

Finally, it would be good to have a table of standard climate metrics for the default version - ECS, TCR, TCRE, ZEC50 and ZEC100.

**We agree, and now provide a new Table 3 with summaries of key metrics.**

Minor issues

Abstract, line 20 - probably too strong to say the model is a robust predictor of 21stC climate until the 21stC is actually over.

**Good point! This has been reworded.**

Line 24 - a zero dimensional model is an extreme interpretation of 'lower resolution'

**We're unclear on this point by the referee, as we don't mention dimensionality, and RCMs are not zero-dimensional. Any clarification would be welcome; thanks.**

Line 63 - similarly, 'well-mixed' is probably redundant if there's no spatial variability.

**Thanks; this has been removed.**

Line 169-169 - missing subscripts
Line 185-187 missing symbols
Line 263-276 this paragraph is repeated

**Thanks for catching these mistakes, which have all been fixed.**

Line 290 - the historical warming not 'observed' in RCMIP
Line 293-296 - similarly, land and carbon sinks in the GCP aren't really observations - they are model results. Fine to make the comparison, but maybe weaken the language about whether the Hector is too strong or weak.

line 295 - too strong to say good historical performance is a general indicator of good climate model performance

**We agree, and have corrected these sentences (lines 295-300).**
* * *
**Referee 4**

This manuscript presents some new developments of the simple (reduced complexity) climate model Hector V3, as well as performances of historical and future climate. As an effective emulator, this model can generally reproduce the global mean temperature and carbon-cycle features of observations and complex climate model simulations.

**We appreciate the overall positive sentiment and thoughtful evaluation.**

As other reviewers have mentioned, besides the references, further detailed and well-structured introductions of the model are expected in section 2.1 and 2.2. (1) For the previous Hector version, it should be better to list the necessary formulas corresponding to the key process of climate change cause-effect chain (i.e., GHG emissions, carbon concentrations, radiative forcing, global warming or temperature change). It's possible not to completely repeat the original form, but to summarize a new expression (like the radiation forcing) with additional information of different GHG components.

**We agree and have worked to provide more detail on the model's parameterization, functionality, and performance; restructure the manuscript for clarity; and discuss Hector's strengths and weaknesses in more depth. In some cases we still fall back on referring readers to previous publications—e.g., for the equations details of the ocean's surface chemistry or how the permafrost thaw was parameterized—but we hope that we have struck a reasonable, and better, balance in this area.**

The featured advantages compared to other models should be highlighted, such as the carbon cycle process of ocean and land.

**In lines 302-312 we now try to emphasize what we see as Hector's primary strengths and reasons for use: it's fully free and open source software; it has enough complexity and flexibility to run unusual experiments such as DACCS; it has unusual features in its terrestrial (e.g., permafrost) and ocean (e.g., surface chemistry) submodels; and its availability as an R package and interactive website greatly broadens the potential audience.**

Additionally, datasets used for forcing and model parameter calibration should be clearly stated, or provide the determined value.

**We have simplified and clarified the calibration description and procedure, making the temperature and CO2 procedures consistent in not holding back any data (although still comparing with other time series not used for calibration), and now clearly give all data sources. See lines 228-242 and Figures 2 and 3.**

And it should be better to present results of Hector coupling with other component models in the Integrated Assessment Model.

**We feel that this is firmly out of scope of the current manuscript; introducing GCAM (Calvin et al. 2019) would vastly change the focus, audience, and complexity of the manuscript.**

(2) For the new model version, it is better to introduce separately the software (kind of user guide) and component process updates (technical report).

**We have tried to separate out software and science-focused updates. That said, the purpose of this manuscript is to document features and performance, not to be a how-to user guide.**

And it is better to provide corresponding tables/figures to demonstrate the impacts of new added or improved processes mentioned in the text, like the permafrost module and carbon tracking.

**Carbon tracking does not change the model behavior or outputs at all; rather, when enabled, extra tracking information is written out, as described by Pressburger et al. (2023). The impact of the permafrost addition was comprehensively described by Woodard et al. (2021) and we think it inadvisable to duplicate that work here; instead, we focus on characterizing the performance of the full v3 model.**

Other Questions and comments:

L465, Figure 1, the dashed lines (numbered 1 and 2) have two arrows, do they mean the two-way exchanges between atmosphere-land and atmosphere-earth? As to the terrestrial carbon cycle processes, they are usually separately depict in vegetation and soils and/or detritus (litter) in most climate models. Are there any duplicate parts between the soil and thawed soil?

**Correct, the two-headed dashed lines are meant to indicate two-way exchanges; we have clarified this in the figure caption. As with most ecosystem and earth system models, vegetation, detritus, and soils are separate pools, as we hope is clear from Figure 1. Soil and thawed soil differ only in that the latter may emit methane to the atmosphere; again, we have clarified this point in the figure caption.**

L479, Table 2, the "2" in cm2/s should be superscripted.

Remove the duplicate information of source, and merge the corresponding table.
L209, the model "Parameterization", may be more appropriate to use "parameter"?

**Thanks for catching these mistakes, which have been corrected.**

More details are needed for the relative parameters and datasets of CMIP complex climate model results used for the Hector model calibration and experiments. For example, how the CMIP6 complex climate model results are used in the paper, multi-model ensemble mean?

**We now provide additional details on these points (259-265).**

And also the key parameters related to temperature increase for both simple and complex climate models, such as the ECS. It is estimated by diagnosing the idealized $CO_2$ sensitivity experiments (in CMIP6 DECK) of complex CSM/ESMs, with significant differences among models. The corresponding part in the last paragraph of section 2.3 should be further refined.

**We have added a new Figure 5 summarizing Hector performance using two CMIP6 DECK protocols, 1%CO2 and 4xCO2, as well as accompanying text in the results (283-293).**

---

## Author Response (AR2)

29 March 2024

Thanks for your consideration of egusphere-2023-1477, "Hector V3.2.0: functionality and performance of a reduced-complexity climate model". We are grateful for the reviewers' making the time to review the manuscript revisions. Our responses below are in **bold**; line numbers refer to the final (clean) version of the revised manuscript.

We hope that the revised manuscript addresses all concerns, but of course welcome further feedback.

Thank you,
Kalyn Dorheim, for the authors
* * *
**Referee 3**

Thanks to the authors, who have worked hard in the revision. The detailed SI is very much appreciated, the calibration documentation is clearer and I'm particularly pleased to see that the default dynamical response of the model to low emission scenarios is now consistent with ESM scenarios. Nice work, and looking forward to seeing more from HECTOR in the future.

**Thank you very much for taking the time to review the manuscript again. Your feedback greatly improved it!**
* * *
**Referee 4**

The paper provides a comprehensive introduction of the Hector model and improvements/updates of the new version (V3.2.0).

The questions and comments:
(1) L550, Figure 1. The solid line of flux (4) representing one-way changes of "the aggregate CO2 from respiration from the terrestrial biosphere and ocean carbon". It should be better to extend the horizontal line to connect the ocean too, i.e., denoting the respirations from both land and ocean.
**Thank you for your suggestion, but we think it is better to leave arrows (4) and (6) separate from one another since they represent very different processes of respiration by the terrestrial biosphere (4) and outgassing from the oceans (6).**

(2) L76, L185-186. There seems to be four temperature components, land (surface), air (troposphere) over land and over the ocean, and sea surface (mixed layer) in Hector. While in the complex CSM/ESM models, the temperatures are generally calculated in the component

models of atmosphere, land, and ocean, etc.. L261-262, Figure 4 (L563) lists three temperatures (air, land and sea surface). The latter two should represent the land surface (not land air) and sea surface temperature and can be compared directly between Hector and ESMs. How about the air temperature? In other words, how to calculate the (surface) air temperature in Hector, average of troposphere temperature over land and ocean?

**We apologize for any confusion. Hector provides a global average air temperature (2 meters above the surface) that is directly comparable to the area-weighted global average of an ESM's 2 m temperature air results. Hector results also include global mean surface temperature which is the area-weighted average of sea surface temperature and land air temperature. The DOECLIM manuscript uses the term tropospheric temperature and air temperature 2 meters above the surface simultaneously.**

**We have clarified the language in our text, removing references to troposphere air temperature (which were only present because of terminology in the previous Kriegler et al. 2005 DOECLIM paper).**

(3) L233, "et al., 2017) et al. (2017)", repeated text.
**Corrected.**

(4) L299, two "reproduces".
**We have changed this wording to clarify the sentence's meaning.**

(5) Supplement L3, SI Table1, too many "radiative forcing" in the description. It's better to omitting the two words or using abbreviation (like RF), and add more information of the GHGs (i.e., black carbon for BC).
**Thank you for the suggestion. We have adopted the abbreviation of RF for radiative forcing, and added more descriptive names for non-halocarbon radiative forcers.**

(6) L35, SI Table6, are the halocarbons in this table should be consistent with those in ST table 1? The CCI4 is not listed in table 1.
**This has been corrected and now there are 26 halocarbons listed in both SI tables 1 and 6.**

(7) L39, SI Table 7, the units "C/yr", missing unit in front of C? And the unit format should be unified through the manuscript, "/yr", or "yr -1".
**We have adopted the $yr^{-1}$ notation and use it consistently throughout both the SI and the main text.**

(8) SI Table 10 and 11, the different ESM results have been used for Hector calculation and comparison. There might cause some inconsistent for result comparisons. Has the author compared the results using the same ESM results in Table 10 and 11?

We appreciate the point and thank the reviewer for raising it, but struggle to see how this is a problem. The fundamental test of the model's future performance is against the combined outputs of the ESMs listed in Table 11, regardless of the source of Hector's parameterization sources (which vary widely). Using the exact same set of models to parameterize the ocean temperature offsets might produce slightly different parameter values, and thus change Hector's tuning, but we're not making any claims about the ESMs' performance. The point of the manuscript—and of model evaluation more generally—is its output relative to observations (for the past) and CMIP (for the future). In summary, we do not believe that this is a problem and prefer to leave the tables as they are.